# A Histogram Analysis of the Pixel Grayscale (Luminous Intensity) of B-Mode Ultrasound Images of the Subcutaneous Layer Predicts the Grade of Leg Edema in Pregnant Women

**DOI:** 10.3390/healthcare11091328

**Published:** 2023-05-05

**Authors:** Eri Ikuta, Masafumi Koshiyama, Yumiko Watanabe, Airi Banba, Nami Yanagisawa, Miwa Nakagawa, Ayumi Ono, Keiko Seki, Haruki Kambe, Taiki Godo, Shin-ichi Sakamoto, Yoko Hara, Akira Nakajima

**Affiliations:** 1Graduate School of Human Nursing, The University of Shiga Prefecture, Hikone 522-8533, Japan; 2Department of Women’s Health, Graduate School of Human Nursing, The University of Shiga Prefecture, Hikone 522-8533, Japan; 3School of Nursing, Tsuruga Nursing University, Tsuruga 914-0814, Japan; 4Department of Electronic Systems Engineering, School of Engineering, The University of Shiga Prefecture, Hikone 522-0057, Japan; 5Iris Women’s Clinic, Hikone 522-0057, Japan

**Keywords:** pixel, grayscale, ultrasound, image, leg edema, photoshop

## Abstract

The technique most widely used to quantitatively measure leg edema is only a pitting edema method. It has recently become possible to digitize B-mode ultrasound images and accurately quantify their brightness using an image-analysis software program. The purpose of this study was to find new indices of the grade of leg skin, to study whether or not analyses of the subcutaneous layer of leg skin on ultrasound images using image-editing software program can be used to evaluate it and to digitize it. Images of 282 subcutaneous layers of leg skin in 141 pregnant women were obtained using a B-scan portable ultrasound device. Rectangular photographs (vertical: skin thickness; horizontal: width of probe) were obtained using an image-editing program, and the luminous intensity (pixel grayscale: 0–255) and thickness of the skin were calculated using a histogram. We investigated the correlation between these parameters and the grade of pitting edema (0–3). There was a significant positive correlation between the grade of pitting edema and the average luminous intensity value, its standard deviation, and the skin thickness (ρ = 0.36, ρ = 0.22, ρ = 0.51, *p* < 0.0001, respectively). In particular, there was strong positive correlation between the grade of pitting edema and both the total number of pixels in a rectangle × (multiplied by) the average luminous intensity value and the total number of pixels in a rectangle × the standard deviation of the average luminous intensity value (ρ = 0.58 and ρ = 0.59, *p* < 0.0001, respectively). We could quantitatively evaluate the grade of leg edema by analyzing ultrasound photographs of the subcutaneous layer of the leg skin using an image-editing software program and found new indices to digitize it.

## 1. Introduction

Leg edema is defined as the accumulation of fluid in the interstitial space due to increased capillary permeability caused by local venous hypertension, systemic venous hypertension, increased plasma volume or decreased plasma oncotic pressure from reduced protein synthesis [1]. Edema is often an early sign of significant fluid retention, which could eventually result in significant cardiac overload and heart failure. Therefore, a reliable and objective method is required for the early diagnosis of leg edema to allow for early treatment and to distinguish physiological edema from more serious conditions.

Some methods to quantitatively measure leg edema have been proposed. Water-displacement volumetry is widely considered to be the reference method for the assessment of edema; however, it is also considered to be time-consuming, difficult to perform, and inappropriate in some clinical situations (preparation of large equipment), which limits the use of this method [2,3]. We previously reported a method using portable ultrasound to quantitatively measure the leg-skin thickness of pregnant women with leg edema [4]. However, we faced a problem in the legs of obese women that showed thick skin, even in cases without edema [5].

The technique most widely used to quantitatively measure leg edema is a subjective clinical assessment, where the examiner applies pressure with his/her index finger to a single location on the patient’s tibia skin (pitting edema method) [4]. An examiner presses firmly with his/her thumb for at least 2 s on each extremity, records the indentation recovery time in seconds, and classifies the grade of leg edema [4,6].

Recently, it has become possible to digitize B-mode ultrasound images and to accurately quantify their brightness with an image-analysis software program [7]. Digitized grayscale images are stored on a computer as a collection of individual spots or pixels of light. A number ranging from 0 (for black) to 255 (for white) is used to represent the brightness of each pixel on the B-mode ultrasound images in an image-editing software program. Thus, previous studies have attempted to analyze the changes in human tissue composition on B-mode ultrasound images [7,8]. For example, a technique using an image-editing software program accurately quantified the human-tissue function of the anterior segment area of the eye [7] and of the transplanted kidney [8].

In the present study, we studied whether or not the analysis of photographs of ultrasound images using an image-editing software program could be used to evaluate the grade of pitting edema of the leg (moisture content). Thus, we compared digitized grayscale images of the subcutaneous layer of leg skin with the results of the pitting edema method. In brief, we investigated the correlation between the pixel grayscale (pixel distribution) of the subcutaneous layer of the leg skin on ultrasound images and the grade of pitting edema, followed by searching for a strongly correlated index of leg edema on the image-editing software program.

## 2. Materials and Methods

### 2.1. Study Subjects and Study Approval

One hundred and forty-one pregnant women (age: 31.9 ± 4.0 years, range: 22–49 years) with 282 legs who attended an outpatient A-clinic in the department of Obstetrics and Gynecology, Hikone City, Shiga, Japan at 36, 37 or 38 weeks of gestation, and who presented with or without lower leg edema (the anterior surface of the tibia), were included in this study. The study protocol was approved by the Ethics Committee of the University of Shiga Prefecture (No. 814), and all pregnant women gave their written informed consent prior to study entry.

### 2.2. Ultrasound Images of the Skin with Leg Edema

Photographs of leg skin in these pregnant women were obtained using a B-scan portable ultrasound device (Viamo sv7, Canon Medical Systems Co., Tochigi, Japan). A 10 MHz ultrasound linear probe was placed on the skin of the lower leg at a definite point, at 6 cm from the upper part of the medial malleolus and 1 cm inside the anterior border of the tibia [4]. This definite point was chosen because the ultrasound technique is easy to apply, as there are few other structures between the skin and bone, and there is a thin fascia. The following anatomic areas were examined: horizontal epidermis, dermis, subcutaneous tissue layer and fascia. The horizontal fascia shows a very high echogenicity. We obtained the longitudinal photographs of these areas along the tibia and cut them into rectangles (vertical: the epidermis, dermis and subcutaneous (bottom was fascia), horizontal: width of probe (37 mm) (Figure 1A,B)). In these photographs, the distance between the skin surface and the upper part of the fascia was measured as the skin thickness, which included the epidermis, dermis and subcutaneous tissue layer [4].

The same trained evaluator (who had been trained for three months) assessed the measured values under the appropriate condition. The light output of the ultrasound device was fixed from the beginning to the end. This is a very important operation.

### 2.3. The Analysis of the Pixel Grayscale (Luminous Intensity) of the Leg Skin

In all cases, rectangular ultrasound photographs of leg skin were created using an image-editing software program, Adobe Photoshop 2021 (Adobe Systems, Inc., San Jose, CA, USA). Longitudinal grayscale images (luminous intensity) of leg skin were opened in Adobe Photoshop 2021. The pixel grayscale of the rectangular ultrasound photograph was measured using the histogram function of Adobe Photoshop (Figure 2A). The histogram represents the grayscale as levels ranging from black (0) to white (255) [9]. In short, the histogram displays one vertical bar for each of 256 brightness levels from black (0) to white (255) in a photograph. The histogram program also shows the histogram data of a photograph [10] (Figure 2B). The total number of pixels are counted in the cropped rectangle of the subcutaneous layer. The mean represents the average pixel grayscale (average luminous intensity value: the average number of pixels in the grayscale image). The standard deviation represents how widely the pixel grayscale (luminous intensity values) is distributed.

### 2.4. The Degree of Pitting Edema of the Lower Leg Skin

At the same time as the photograph of the lower leg skin was obtained, a qualitative evaluation of the degree of physiological pitting edema of the skin at the same site was performed. Finger pressure was applied to the swollen site of the skin to determine whether an indentation formed that persisted after pressure was removed. The grades of pitting edema were as follows: Grade 0, negative for edema, with no persisting indentation after the release of finger pressure; Grade 1, mild pitting edema that disappeared within 10 s; Grade 2, moderate pitting edema that disappeared after 10–15 s; and Grade 3, severe pitting edema that lasted for more than 15 s after the release of finger pressure [4].

### 2.5. Statistical Analysis

All statistical analyses were performed using the JMP Pro software program, (version 14, SAS Institute Japan, Tokyo, Japan). Correlations between the parameters (total pixels, mean, standard deviation, skin thickness, total pixels × (multiplied by) average and total pixels × standard deviation) and the grade of pitting edema were calculated using a rank correlation analysis and compared using Spearman’s rank correlation coefficient. *p*-values of <0.05 were considered statistically significant.

## 3. Results

### 3.1. The Grades of Lower Leg Skin Edema

The grades of edema were as follows: Grade 0 (n = 75), Grade 1 (n = 110), Grade 2 (n = 59) and Grade 3 (n = 38). In all cases, edema was diagnosed in the pregnant women and was tested twice.

### 3.2. The Relationship between the Lower Leg Skin Data in the Histogram of the Image Editing Software Program and the Grade of Pitting Edema

Using Spearman’s correlation of the ranked values of the data, a significant positive correlation was observed between the grade of pitting edema and the total number of pixels in the rectangle of the subcutaneous layer of lower leg skin (ρ = 0.51, *p* < 0.0001) (Figure 3A). A significant positive correlation was also observed between the grade of pitting edema and the average pixel grayscale (average luminous intensity value), standard deviation of the pixel grayscale, and skin thickness (ρ = 0.36, ρ = 0.22, ρ = 0.51, *p* < 0.0001, respectively) (Figure 3B–D). In particular, there was a strong positive correlation between the grade of pitting edema and both the total number of pixels in the rectangle × the average pixel grayscale and the total number of pixels in the rectangle × standard deviation of the pixel grayscale (ρ = 0.58 and ρ = 0.59, *p* < 0.0001, respectively) (Figure 4A,B). A significant positive correlation was observed between the grade of pitting edema and the total number of pixels, the average pixel grayscale, and the standard deviation of the pixel grayscale, respectively.

In order to detect a higher correlation coefficient, we furthermore calculated the total number of pixels × the average pixel grayscale and the total number of pixels × standard deviation of the pixel grayscale.

We can note that increased leg edema showed an increased skin thickness and high echogenicity as a whole (Figure 5A,B).

## 4. Discussion

In the clinical setting, there exists only a pitting edema method to quantitatively measure leg edema. This method needs more than a few minutes and involves operator variability. The purpose of this study was to find a new index of the grade of leg skin edema in order to measure it in a short time.

A portable ultrasound device is a reliable, objective, and quantitative method for identifying legs with and without leg edema and does not include operator variability [4]. The ultrasound device is portable, and the imaging method is easy to use, produces consistent results between operators and detects leg skin with constant brightness in black and white. A doctor or nurse would easily be able to perform this imaging method in an outpatient setting.

We previously showed that a portable B-scan ultrasound system could be used to quantitatively measure the increased thickness of the skin of the leg in pregnant women with leg edema [4]. In that report, the correlation coefficient (ρ) was 0.56 (*p* < 0.0001) and was close to the correlation coefficient of the present study (ρ = 0.51). However, we faced a problem in that the legs of obese women showed thick skin, even in cases without edema [5]. The measurement of the skin thickness is useful as a quantitative judgement of the effects of treatment alone for women who are diagnosed with edema in advance [11]. Thus, we searched for an index that can be used for both non-obese and obese pregnant women with leg edema. The digitized grayscale images of leg skin in the present study could be used for the evaluation of both non-obese and obese pregnant women.

When leg edema worsens, a greater amount of fluid collects in the subcutaneous layer. Thus, it causes the increased thickness of the skin leg. The increased thickness of the skin entails the increased gross area of the cropped rectangular. In the present study, a measurement of skin thickness using a portable ultrasound device was performed after pitting edema disappearance. A significant positive correlation was observed in the grade of pitting edema and both skin thickness (Figure 3D) and the total number of pixels in the rectangle (Figure 3A).

In the present study, increased leg edema was positively correlated with an increased brightness of the pixel grayscale, increased standard deviation of the pixel grayscale, and increased skin thickness. Generally, the echogenicity of water is very low (black), while that of hard tissue, such as bones, is very high. Why does increased leg skin edema show increased brightness? We hypothesize that the adipose tissue in the subcutaneous tissue shows an increased echogenicity and brightness on the grayscale when increasing water. Strictly speaking, water shows a very low echogenicity and decreased brightness. These phenomena are seemingly contradictory. We hypothesize that the increased echogenicity of adipose tissue goes beyond the low echogenicity of water. Adipose tissue might become firm and its acoustic impedance might increase due to the pressure of increased fluid. In fact, we observed a strong positive correlation between the grade of pitting edema and the pixels number × the average pixel grayscale (ρ = 0.58).

Suehiro et al. reported that the subcutaneous echogenicity grade was higher in the upper medial leg and the lower medial/lateral leg than in the upper medial thigh in dependent edema [12]. In dependent edema, the lower the leg level, the higher the echogenicity. They noted that this finding might indicate that this distribution simply follows gravity. Their data were not inconsistent with our findings that an increased edema grade was associated with an increased brightness level on the histogram. They also pointed out that the rate of subcutaneous echo-free space was higher in all parts of the leg than in the upper medial thigh in dependent edema. The structural changes on ultrasound images of legs with class C3 edema (according to the clinical-etiology-anatomy-pathophysiology [CEAP] classification) with venous disease reflected homogeneous subcutaneous layer thickening [13]. Caggiati advocated that two main ultrasound patterns can be observed: (1) the homogeneous thickening of the subcutaneous layer with a slight increase of echogenicity; and (2) the presence of anechoic lacunae that are greatly variable in size, extension and orientation. These findings were also found in our study. In the present study, we also observed a strong positive correlation between the edema grade and total number of pixels in the rectangle × standard deviation of luminous intensity (ρ = 0.59). As the grade of edema grows higher, the variation of the pixel grayscale may become wider. This may mean that anechoic lacunae (low echogenicity) develop sporadically in the subcutaneous layer when edema becomes severe. Our data were based on all selected pregnant women with or without leg edema. Regardless of age, sex, and weight, the same might be true of other leg edemas (only C3 edemas) due to water leakage, except for lymphedema, ulcer formation and sclerosis.

Photoshop is a widely-used software program that provides simple, effective, and time-saving methods for two-dimensional image processing. Recently, the analysis of grayscale ultrasound image files has also been used for the diagnosis of disease. Imaging analyses were reported to accurately quantify carotid plaque, intraplaque hemorrhage, fibromuscular tissue, calcium and lipid [14,15]. Lal B.K. et al. reported that the median grayscale intensity (range) in control subjects was 2 (0 to 4) for blood, 12 (8 to 26) for lipid, 53 (41 to 76) for muscle, 172 (112 to 196) for fibrous tissue, and 221 (211 to 255) for calcium [15]. They concluded that computer-assisted pixel distribution analysis of duplex ultrasound scan images accurately quantified intraplaque hemorrhage, fibromuscular tissue, calcium, and lipid. Basically, the harder the material, the higher the grayscale level. This technique was also used to diagnose intraretinal fluid turbidity in patients with diabetic macular edema [16], differences between malignant and benign thyroid nodule tissues [17], intrinsic plantar muscle differences between hemiparesis and contralateral feet in post-stroke patients [18], and infantile hemangiomas [19]. Calvo-Lobo C. et al. showed that image J software differences in B-mode ultrasound imaging with a reduction of the abductor hallucis pixels count were presented between hemiparetic and contralateral feet in poststroke patients [18]. They made no mention of the reason for this. Recently, Soni NJ et al. reported on pleural fluid echogenicity measured by ultrasound image pixel density to differentiate transudative versus exudative pleural effusions [20]. They concluded that the echogenicity was significantly higher in exudative versus transudative pleural effusions. The correlation of the pleural-fluid pixel density with LDH and protein was seen in exudative effusions. This phenomenon might occur due to an increase of the acoustic impedance of pleural fluid. However, the true principles are not clear, and the analyses are still restrictive.

In the future, we should develop a method that allows for the simultaneous qualitative and quantitative evaluation of leg edema using the present technique. Thus, we would only take an ultrasound photograph of leg skin, and it will be possible for us to perform an automatic measurement of the grade of leg skin edema immediately.

## 5. Conclusions

As a new method, we were able to analyze ultrasound photographs of leg skin using an image-editing software program to quantitatively evaluate the grade of leg edema.

## Figures and Tables

**Figure 1 healthcare-11-01328-f001:**
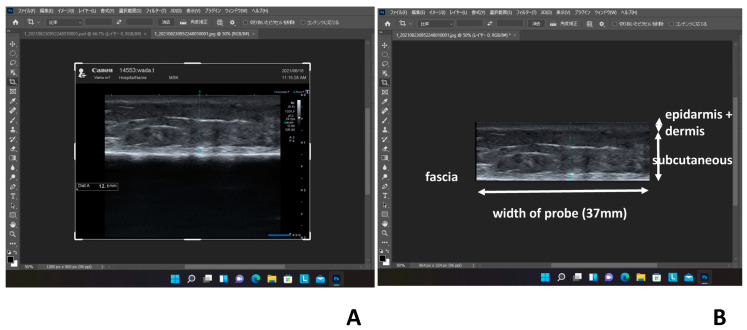
(**A**) We obtained a longitudinal photograph of leg skin (horizontal epidermis, dermis, subcutaneous tissue layer and fascia). (**B**) In all cases, rectangular ultrasound photographs of leg skin were created using an image-editing software program.

**Figure 2 healthcare-11-01328-f002:**
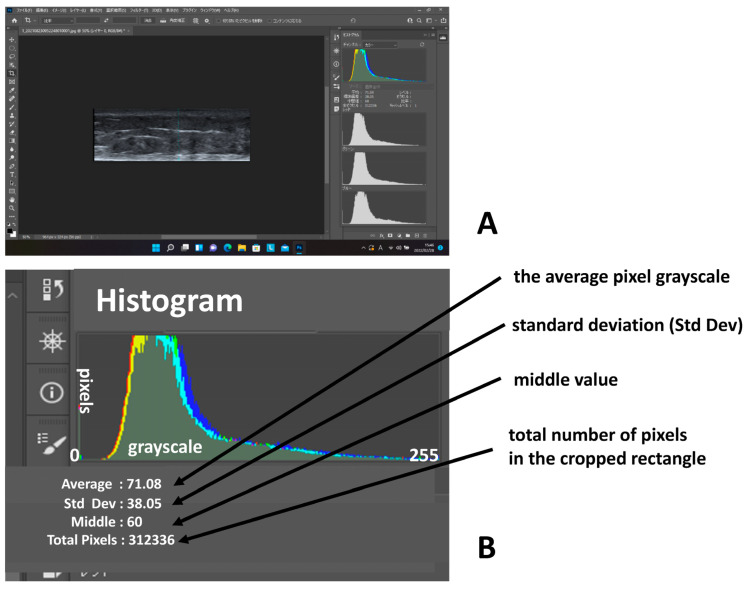
(**A**) The pixel grayscale of the rectangular ultrasound photograph was measured using the histogram function of Adobe Photoshop. (**B**) The histogram program also showed the histogram data of a photograph. (The total number of pixels were counted in the cropped rectangle of the subcutaneous layer. The average pixel grayscale represented the average of the histogram. The standard deviation represented the width of the histogram.).

**Figure 3 healthcare-11-01328-f003:**
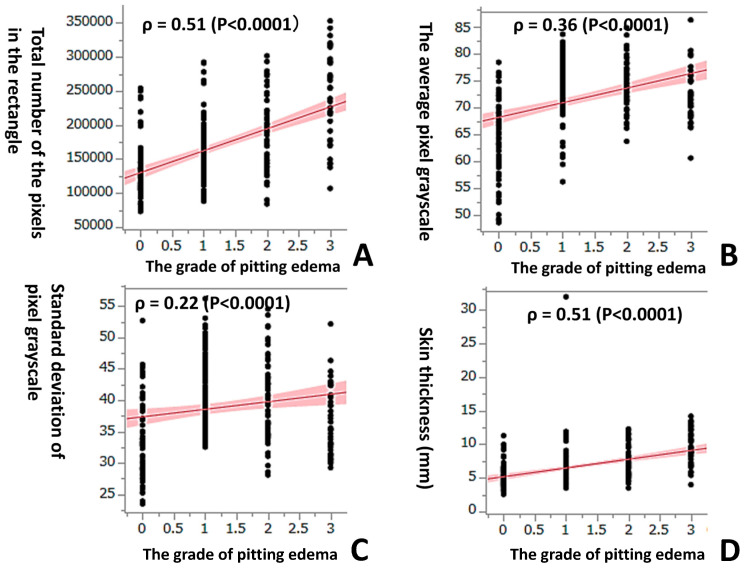
(**A**) Using Spearman’s correlation of the ranked values of the data, a significant positive correlation was observed between the grade of pitting edema and the total number of pixels in the rectangle. A significant positive correlation was also observed between the grade of pitting edema and (**B**) the average pixel grayscale, (**C**) standard deviation of the pixel grayscale, and (**D**) skin thickness.

**Figure 4 healthcare-11-01328-f004:**
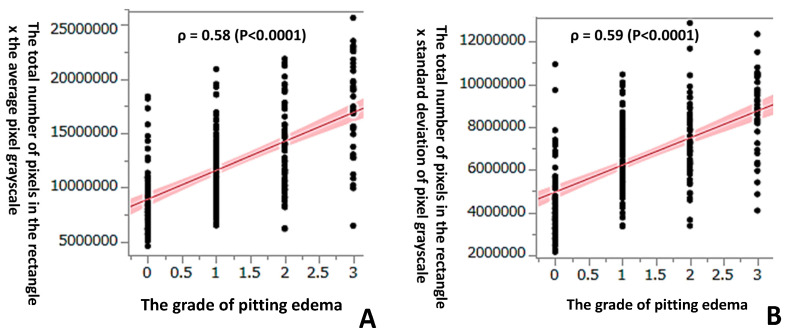
(**A**) There was a strong positive correlation between the grade of pitting edema and both the total number of pixels in the rectangle × the average pixel grayscale and (**B**) the total number of pixels in the rectangle × standard deviation of the pixel grayscale.

**Figure 5 healthcare-11-01328-f005:**
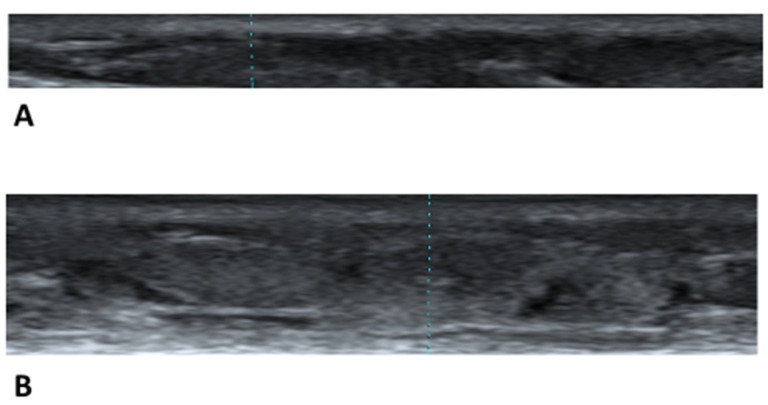
(**A**) Typical ultrasound image of the leg skin without edema. This shows the skin thickness: 3.8 mm; the total number of pixels: 109,668: the average pixel grayscale: 53.62; and the standard deviation: 29.62. (**B**) Typical ultrasound image of Grade 3 (severe) pitting edema. This shows the skin thickness: 8.2 mm; the total number of pixels: 205,868; the average pixel grayscale: 75.98; and the standard deviation: 43.03.

## Data Availability

Not applicable.

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
