# Peer review of "A Histogram Analysis of the Pixel Grayscale (Luminous Intensity) of B-Mode Ultrasound Images of the Subcutaneous Layer Predicts the Grade of Leg Edema in Pregnant Women"

_healthcare, 2023, doi:10.3390/healthcare11091328_

Round 1
Reviewer 1 Report
This is a very interesting manuscript title. The abstract needs some refinement. It is written too generally. The author does not specify what the purpose of the work is or what the result is. The strength of the work is the methodology. Poor, however, is a very poor review of the literature, which significantly reduces the value of the work. In its current form, in my opinion, the article is not suitable for publication. The summary is also not refined, especially in relation to the purpose of the study once it has been conducted.
The main research contributions are not clear. The author needs to write the main research contributions in bullet points with technical proof
Unclear problem statement and discussion of core contribution.
The authors are recommended to improve the literature survey with some recent works.
- Discuss literature reviews based on their performance metrics along with the boons and limitations of each work surveyed.
Discuss the reasons why the proposed work is offering better results in comparative analysis with the other similar methods.
- What are the strongest and weakest aspects of this work?
- About the datasets used by the authors, it is not clear if they are publicly available, otherwise (if they are private datasets) the possibility of reproducing the experiments, an indispensable aspect in any scientific work, would not be guaranteed, then the authors should clarify this aspect;
- The "Conclusions and Future Work" section appears excessively concise, then the authors should expand it by recapping all the steps to the proposed work, as to offer a brief but complete summary of it to the readers.
I suggest that the authors introduce certain taxonomy, at least through subsections.
Author Response
We corrected our article with red color in Text.
The abstract needs some refinement. It is written too generally. The author does not specify what the purpose of the work is or what the result is.
(A)>>Thank you for your good advices. We added “The technique most widely used to quantitatively measure leg edema is only a pitting edema method..” and “The purpose of this study was to find new indices of the grade of leg skin, to study whether or not analyses of the subcutaneous layer of leg skin on ultrasound images using image-editing software program can be used to evaluate it and to digitize it..”and “We could quantitatively evaluate the grade of leg edema by analyzing ultrasound photographs of the subcutaneous layer of the leg skin using an image editing software program and found new indices to digitize it.” in Abstract.
The strength of the work is the methodology. Poor, however, is a very poor review of the literature, which significantly reduces the value of the work. In its current form, in my opinion, the article is not suitable for publication. The summary is also not refined, especially in relation to the purpose of the study once it has been conducted.
- èThank you for good advice. We added “ There is only a pitting edema method to
quantitatively measure leg edema in the clinical setting. This method needs more than a few minutes and involves operator variability. The purpose of this study was to find a new index of
the grade of leg skin in order to measure it in a short time.
Portable ultrasound device is a reliable, objective, and quantitative method for identifying legs with and without leg edema and is without operator variability [4]. The ultrasound device is portable, the imaging method is easy to use, produces consistent results between operators and detects leg skin with constant brightness in black and white. A doctor or nurse would be able to perform this imaging method easily in an outpatient setting.
When leg edema worsens, a greater amount of fluid collects in the subcutaneous
layer. Thus, it causes the increased thickness of the skin leg. The increased thickness of
the skin means the increased gross area of the cropped rectangular. In the present study, measurement of skin thickness using a portable ultrasound device was performed after
pitting edema disappearance. A significant positive correlation was observed the grade
of pitting edema and skin thickness (Figure 3D), and the total number of pixels in the
rectangle (Figure 3A). in Discussion.
Thus, it will be possible for us to perform the automatic measurement of the grade of leg skin edema immediately, if we only take an ultrasound photograph of leg skin. in Discussion.
We were able to analyze ultrasound photographs of leg skin using an image editing software program to quantitatively evaluate the grade of leg edema as a new method. In Conclusion.
The main research contributions are not clear. The author needs to write the main research contributions in bullet points with technical proof
(A)>>Our research contribution involves in the following sentences.: In the future, we should develop a method that allows the simultaneous qualitative and quantitative evaluation of leg edema using the present technique. Thus, it will be possible for us to perform the automatic measurement of the grade of leg skin edema immediately, if we only take an ultrasound photograph of leg skin. in Discussion.
Unclear problem statement and discussion of core contribution.
(A)è>>The present problem and core contribution is following :There is only a pitting edema method to quantitatively measure leg edema in the clinical setting. This method needs more than a few minutes and involves operator variability. The purpose of this study was to find a new index of the grade of leg skin in order to measure it in a short time.
Our research contribution involves in the following sentences.: In the future, we should develop a method that allows the simultaneous qualitative and quantitative evaluation of leg edema using the present technique. Thus, it will be possible for us to perform the automatic measurement of the grade of leg skin edema immediately, if we only take an ultrasound photograph of leg skin. in Discussion.
The authors are recommended to improve the literature survey with some recent works.
- Discuss literature reviews based on their performance metrics along with the boons and limitations of each work surveyed.
(A)>>We added the following sentences :Lai BK et al. reported that the median grayscale intensity (range) in control subjects was 2 (0 to 4) for blood, 12 (8 to 26) for lipid, 53 (41 to 76) for muscle, 172 (112 to 196) for fibrous tissue, and 221 (211 to 255) for calcium [15]. They concluded that computer-assisted pixel distribution analysis of duplex ultrasound scan images accurately quantified intraplaque hemorrhage, fibromuscular tissue, calcium, and lipid. Basically, the harder the material, the higher the grayscale level. in Discussion.
Calvo-Lobo C. et al. showed that image J software differences in B-mode ultrasound imaging with a reduction of the abductor hallucis pixels count were presented between hemiparetic and contralateral feet in poststroke patients [18]. They made no mention of its reason. Recently, Soni NJ et al. reported about pleural fluid echogenicity measured by ultrasound image pixel density to differentiate transudative versus exudative pleural effusions [20]. They concluded that its echogenicity was significantly higher in exudative versus transudative pleural effusions. Correlation of pleural fluid pixel density versus LDH and protein was seen in exudative effusions. This phenomenon might occur due to increase of acoustic impedance of pleural fluid. However, true principles are not clear and the analyses are still restrictive. in Discussion.
Discuss the reasons why the proposed work is offering better results in comparative analysis with the other similar methods.
- What are the strongest and weakest aspects of this work?
(A)>>the strong aspect: Our work will lead to the following in the future: We only take an ultrasound photograph of leg skin and it will be possible for us to perform the automatic measurement of the grade of leg skin edema immediately.
the weakest aspect: We hypothesize that the increased echogenicity of adipose tissue goes beyond low echogenicity of water. Adipose tissue might become firm and its acoustic impedance increases, due to pressure of increased fluid. However, true principles are not clear and the analyses are still restrictive.
- About the datasets used by the authors, it is not clear if they are publicly available, otherwise (if they are private datasets) the possibility of reproducing the experiments, an indispensable aspect in any scientific work, would not be guaranteed, then the authors should clarify this aspect;
(A)>>Thank you. We can present our datasets if the reader have any question or request.
The "Conclusions and Future Work" section appears excessively concise, then the authors should expand it by recapping all the steps to the proposed work, as to offer a brief but complete summary of it to the readers.
(A)>>Thank you. We changed to: In the future, we should develop a method that allows the simultaneous qualitative and quantitative evaluation of leg edema using the present technique. Thus, we only take an ultrasound photograph of leg skin and it will be possible for us to perform the automatic measurement of the grade of leg skin edema immediately.
- Conclusion
We were able to analyze ultrasound photographs of leg skin using an image editing software program to quantitatively evaluate the grade of leg edema as a new method.
I suggest that the authors introduce certain taxonomy, at least through subsections.
 (A)>>We cannot understand the meaning of the referee’s certain taxonomy.
The present study is to find new indices of the grade of leg skin using B-mode ultrasound images. It’s taxonomy of the present study.
Reviewer 2 Report
Thank you for submitting your report on the correlation between leg edema grade and subcutaneous layer images using pixel grayscale analysis. The report is well-written and informative, and the authors have made progress in evaluating leg edema grades using image analysis. However, there are still a few questions that need to be addressed before the manuscript can be reconsidered. Please consider the following points:
1. In Fig.3A, it is unclear why the total number of pixels increases with the increasing grade of pitting edema. If severe pitting edema lasts, then the area of the cropped rectangular image should be smaller than the recovered skin image.
In another way, edema means more water in the tissue (which has low echogenicity), resulting in a lower grayscale value. However, as Fig.3 shows, the grayscale increases with an increased grade of edema.
Please provide an explanation for this in the manuscript.
2. In Fig. 3D, it is counterintuitive that skin thickness increases with the grade of pitting edema, as persistent indentation would suggest the opposite. Please provide a discussion of this phenomenon for readers. Additionally, please clarify when and how you measure skin thickness (after or before pitting edema disappearance).
3. Regarding Fig.3, please clarify how you plotted the lines with the red shadow. Are they fitted with the average value of discrete black points? Also, please explain the meaning of the red shadow.
4. In Fig. 4A, it is not clear what the meaning of "total number of pixels in the rectangle * the average pixel grayscale" is, and why you chose to plot this. The same question applies to Fig. 4B. Please provide an explanation in the manuscript to help readers understand better.
5. In line 170, you mention that digitized grayscale images can be used for the evaluation of both non-obese and obese pregnant women. Please clarify how digitized grayscale images solve the problem due to obesity.
Additionally, please consider the following minor points:
1. The content of the abstract is clear, but it would be better to remove "Background," "Methods," "Results," and "Conclusion" when the manuscript is published.
2. Please avoid repeating text. For example, the second sentence in the caption of Fig. 1B and the first sentence in section 2.3 are identical. Every sentence should provide new information.
3. In the caption of Fig.2A, "The average pixel greyscale represented the average pixel grayscale" does not provide any explanation. Please rephrase this sentence.
4. In Fig. 2B, the caption "The standard deviation represented how widely the pixel grayscale was distributed" is identical to the sentence in lines 110-111. Please avoid using duplicate text.
Thank you for your attention to these points. We look forward to receiving your revised manuscript.
Author Response
We corrected our article with red color in Text.
- In Fig.3A, it is unclear why the total number of pixels increases with the increasing grade of pitting edema. If severe pitting edema lasts, then the area of the cropped rectangular image should be smaller than the recovered skin image.
In another way, edema means more water in the tissue (which has low echogenicity), resulting in a lower grayscale value. However, as Fig.3 shows, the grayscale increases with an increased grade of edema.
Please provide an explanation for this in the manuscript.
(A)>>f severe pitting edema lasts, the adipose tissue in the subcutaneous tissue shows increased echogenicity and the thickness of the subcutaneous layer of the leg is more significantly increased that means the gross area of the increased cropped rectangular.
And we added the following sentence in Discussion. :
We hypothesize that the increased echogenicity of adipose tissue goes beyond low echogenicity of water. Adipose tissue might become firm and its acoustic impedance increases, due to pressure of increased fluid.
- In Fig. 3D, it is counterintuitive that skin thickness increases with the grade of pitting edema, as persistent indentation would suggest the opposite. Please provide a discussion of this phenomenon for readers. Additionally, please clarify when and how you measure skin thickness (after or before pitting edema disappearance).
(A)>>Yes. We added the following sentence in Discussion: When leg edema wosens, a greater amount of fluid collects in the subcutaneous layer. Thus, it causes the increased thickness of the skin leg. The increased thickness of the skin means the increased gross area of the cropped rectangular. In the present study, measurement of skin thickness using a portable ultrasound device was performed after pitting edema disappearance. A significant positive correlation was observed the grade of pitting edema and skin thickness (Figure 3D), and the total number of pixels in the rectangle (Figure 3A).
- Regarding Fig.3, please clarify how you plotted the lines with the red shadow. Are they fitted with the average value of discrete black points? Also, please explain the meaning of the red shadow.
(A)>>Yes. It is the average value of discrete black points. The red line is set in the statistics soft
- In Fig. 4A, it is not clear what the meaning of "total number of pixels in the rectangle * the average pixel grayscale" is, and why you chose to plot this. The same question applies to Fig. 4B. Please provide an explanation in the manuscript to help readers understand better.
(A)>>The total number of pixels in the rectangle multiplied by the average pixel grayscale Yes, we added the following sentences in Results: The significant positive correlation was observed between the grade of pitting edema and the total number of the pixels, the average pixel grayscale, and standard deviation of pixel grayscale, respectively.
In order to detect higher correlation coefficient, furthermore, we calculated the total number of
pixels × the average pixel grayscale and the total number of pixels in × standard deviation of pixel grayscale.
.
- In line 170, you mention that digitized grayscale images can be used for the evaluation of both non-obese and obese pregnant women. Please clarify how digitized grayscale images solve the problem due to obesity.
(A)>>In additional analysis, we can calculate cut off value regardless of non-obese and obese.
It means that digitized grayscale images of fluid collect is similar tendency regardless of non-obese and obese.
Additionally, please consider the following minor points:
- The content of the abstract is clear, but it would be better to remove "Background," "Methods," "Results," and "Conclusion" when the manuscript is published.
(A)>>Thank you for your advice. We remove "Background," "Methods," "Results," and "Conclusion" in Abstract.
- Please avoid repeating text. For example, the second sentence in the caption of Fig. 1B and the first sentence in section 2.3 are identical. Every sentence should provide new information.
(A)>>Thank you for your advice. We remove the following sentence in Fig.1B: (B) We cut this photograph into rectangles [vertical: the epidermis, dermis and subcutaneous (bottom was fascia), horizontal: width of probe (37mm).
- In the caption of Fig.2A, "The average pixel greyscale represented the average pixel grayscale" does not provide any explanation. Please rephrase this sentence.
(A)>>Yes! We changed it to “The average pixel grayscale represented the average of the histogram.”
- In Fig. 2B, the caption "The standard deviation represented how widely the pixel grayscale was distributed" is identical to the sentence in lines 110-111. Please avoid using duplicate text.
(A)>>Yes! We changed it to “The standard deviation represented width of the histogram.”
Reviewer 3 Report
The study aimed to determine whether analyzing ultrasound images of the subcutaneous layer of leg skin using image-editing software can be used to evaluate the grade of leg edema. It was concluded that the use of an image editing software program can be used to quantitatively evaluate the grade of leg edema. I recommend rejection of the paper due to the following reasons:
While it is true that digitizing B-mode ultrasound images and accurately quantifying their brightness is possible, using raw pixel values between 0 and 255 to detect a disease is a very bad way of doing so. These numbers are meaningless on their own and need to be normalized for reliable analysis. Raw pixel values only provide information about the brightness of each individual pixel and do not take into account the overall structure or features of the image, which are crucial for accurate diagnosis.
Using just the raw brightness of pixels to detect a disease is a very naive approach and can lead to inaccurate results. For example, if there is a variation in the lighting conditions or imaging equipment, it can lead to variations in the raw pixel values, which can then lead to incorrect diagnoses. Normalizing the raw pixel values, or even better, using more advanced techniques such as machine learning and computer vision, can help to mitigate these issues and provide more accurate results.
There are much better and more advanced techniques available, such as machine learning and computer vision, that can be used to accurately detect diseases and abnormalities in medical images. These techniques are designed to identify patterns and features in medical images that are indicative of a particular disease, rather than just relying on the raw brightness of individual pixels. By training a machine learning model on a large dataset of medical images, it can learn to identify these patterns and features with high accuracy, even in cases where the human eye may not be able to see them.
Additionally, the paper's writing could benefit from further clarification and coherence to ensure that its findings and implications are clearly communicated to the reader.
Based on these concerns, I recommend rejecting the paper in its current form. However, I encourage the authors to revise their study by addressing these issues and incorporating more advanced techniques and approaches.
Regards,
Author Response
We corrected our article with red color in Text.
While it is true that digitizing B-mode ultrasound images and accurately quantifying their brightness is possible, using raw pixel values between 0 and 255 to detect a disease is a very bad way of doing so. These numbers are meaningless on their own and need to be normalized for reliable analysis. Raw pixel values only provide information about the brightness of each individual pixel and do not take into account the overall structure or features of the image, which are crucial for accurate diagnosis.
(A)>>Thank you for your comments. It has been reported that edematous skin shows increased echogenicity [12, 13]. The increased echogenicity of adipose tissue in subcutaneous layer goes beyond low echogenicity of water. Adipose tissue might become firm and its acoustic impedance increases, due to pressure of increased fluid. In a case of light edema, the subcutaneous layer shows homogeneous thickening [13]. Thus, grayscale (brightness) can be measured considering fluid content.
Using just the raw brightness of pixels to detect a disease is a very naive approach and can lead to inaccurate results. For example, if there is a variation in the lighting conditions or imaging equipment, it can lead to variations in the raw pixel values, which can then lead to incorrect diagnoses. Normalizing the raw pixel values, or even better, using more advanced techniques such as machine learning and computer vision, can help to mitigate these issues and provide more accurate results.
(A)>>We are sorry. We measured the leg skin using the same ultrasound device. It records the subcutaneous layer with constant brightness in black and white. We added the following sentence: In the future, we should develop a method that allows the simultaneous qualitative and quantitative evaluation of leg edema using the present technique. Thus, we only take an ultrasound photograph of leg skin and it will be possible for us to perform the automatic measurement of the grade of leg skin edema immediately.
There are much better and more advanced techniques available, such as machine learning and computer vision, that can be used to accurately detect diseases and abnormalities in medical images. These techniques are designed to identify patterns and features in medical images that are indicative of a particular disease, rather than just relying on the raw brightness of individual pixels. By training a machine learning model on a large dataset of medical images, it can learn to identify these patterns and features with high accuracy, even in cases where the human eye may not be able to see them.
(A)>>Yes, we can diagnose disease more accurately using AI. But, on a daily basis, we use ultrasound device in medicine. Thus, there is meaning in taking advantage of ultrasound photographs. Thus, we analyzed these photographs to quantify leg edema. We added the following sentence in Discussion:
There is only a pitting edema method to quantitatively measure leg edema in the clinical setting. This method needs more than a few minutes and involves operator variability. The purpose of this study was to find a new index of the grade of leg skin edema in order to measure it in a short time.Portable ultrasound device is a reliable, objective, and quantitative method for identifying legs with and without leg edema and is without operator variability [4]. The ultrasound device is portable, the imaging method is easy to use, produces consistent results between operators and detects leg skin with constant brightness in black and white. A doctor or nurse would be able to perform this imaging method easily in an outpatient setting.
Additionally, the paper's writing could benefit from further clarification and coherence to ensure that its findings and implications are clearly communicated to the reader.
Based on these concerns, I recommend rejecting the paper in its current form. However, I encourage the authors to revise their study by addressing these issues and incorporating more advanced techniques and approaches.
Regards,
(A)>>Thank you for your advice. There were some reports analyzing grayscale of ultrasound photograph to diagnose disease [14-20]. One of our authors is Dr.Sakamoto, who is a electronic systems engineering. He checked our technique using grayscale of ultrasound photograph and said no problem.
Then, we added the following sentences in Discussion.:
Lai BK et al. reported that the median grayscale intensity (range) in control subjects was 2 (0 to 4) for blood, 12 (8 to 26) for lipid, 53 (41 to 76) for muscle, 172 (112 to 196) for fibrous tissue, and 221 (211 to 255) for calcium [15]. They concluded that computer-assisted pixel distribution analysis of duplex ultrasound scan images accurately quantified intraplaque hemorrhage, fibromuscular tissue, calcium, and lipid. Basically, the harder the material, the higher the grayscale level.
Calvo-Lobo C. et al. showed that image J software differences in B-mode ultrasound imaging with a reduction of the abductor hallucis pixels count were presented between hemiparetic and contralateral feet in poststroke patients [18]. They made no mention of its reason. Recently, Soni NJ et al. reported about pleural fluid echogenicity measured by ultrasound image pixel density to differentiate transudative versus exudative pleural effusions [20]. They concluded that its echogenicity was significantly higher in exudative versus transudative pleural effusions. Correlation of pleural fluid pixel density versus LDH and protein was seen in exudative effusions. This phenomenon might occur due to increase of acoustic impedance of pleural fluid. However, true principles are not clear and the analyses are still restrictive. 
In the future, we should develop a method that allows the simultaneous qualitative and quantitative evaluation of leg edema using the present technique. Thus, we only take an ultrasound photograph of leg skin and it will be possible for us to perform the automatic measurement of the grade of leg skin edema immediately.
Why does increased leg skin edema show increased brightness? We hypothesize that the adipose tissue in the subcutaneous tissue shows increased echogenicity and brightness on grayscale with increasing water. Strictly, water shows very low echogenicity and decreased brightness. These phenomena are seemingly contradictory. We hypothesize that the increased echogenicity of adipose tissue goes beyond low echogenicity of water. Adipose tissue might become firm and its acoustic impedance increases, due to pressure of increased fluid.
Portable ultrasound device is a reliable, objective, and quantitative method for identifying legs with and without leg edema and is without operator variability [4]. The ultrasound device is portable, the imaging method is easy to use, produces consistent results between operators and detects leg skin with constant brightness in black and white.A doctor or nurse would be able to perform this imaging method easily in an outpatient setting.
Round 2
Reviewer 1 Report
- I think that all of my review' criticisms have been answered and the paper is now ready for acceptance.
Author Response
I think that all of my review' criticisms have been answered and the paper is now ready for acceptance.
>> (A) Thank you very much for your kind desion.
Reviewer 3 Report
I persist in recommending rejection of the paper, as the methodology exhibits critical shortcomings stemming from a lack of innovation and scientific rigor.
This methodology is problematic primarily because it overlooks the complexities of ultrasound image interpretation and the inherent variability in raw brightness values. By exclusively relying on raw brightness, the authors disregard essential image features and contextual information that could provide a more accurate diagnosis of leg skin edema. Ultrasound images inherently contain speckle noise and artifacts, which can significantly impact the raw brightness values and lead to false-positive or false-negative diagnoses. Moreover, factors such as the angle of insonation, transducer pressure, and variations in tissue composition can also contribute to inconsistencies in raw brightness levels, further complicating the diagnostic process.
As well, numerous environmental and software factors, such as ISO control and ambient lighting, can influence brightness levels. Relying on this data for diagnoses is overly simplistic and could prove dangerous.
In addition to the aforementioned concerns, the current methodology's simplicity also limits its applicability and adaptability to different clinical scenarios and patient populations. Leg skin edema can result from various underlying conditions, such as venous insufficiency, lymphedema, or heart failure, which may present with distinct ultrasound image characteristics. A diagnostic method that only considers raw brightness values lacks the capability to differentiate between these conditions or account for patient-specific factors that could influence image interpretation, such as age, body mass index, or comorbidities.
Furthermore, the authors' claim that "One of our authors is Dr.Sakamoto, who is a electronic systems engineering. He checked our technique using grayscale of ultrasound photograph and said no problem." does not provide a satisfactory resolution to the concerns I raised. This statement constitutes an unsound appeal to authority instead of a well-reasoned scientific response.
Author Response
Comments and Suggestions for Authors
I persist in recommending rejection of the paper, as the methodology exhibits critical shortcomings stemming from a lack of innovation and scientific rigor.
This methodology is problematic primarily because it overlooks the complexities of ultrasound image interpretation and the inherent variability in raw brightness values. By exclusively relying on raw brightness, the authors disregard essential image features and contextual information that could provide a more accurate diagnosis of leg skin edema. Ultrasound images inherently contain speckle noise and artifacts, which can significantly impact the raw brightness values and lead to false-positive or false-negative diagnoses.
>> (A) Thank you for good advice. The subcutaneou layer contains adipose tissue. The ultrasound image of it’s layer without edema shows diffuse low echogenicity. When edema occurs, fluid leaks almost diffusely. Increased edema grade, however, shows increased high echogenicity. Thus, we added Figure 5 (A)(B). There were two figures. One was typical leg skin tissue without edema and another was typical severe edema tissue. We added the following sentence in Results: We can notice that increased leg edema showed increased skin thickness and high echogenicity as a whole. As you point out, speckle noise and artifacts are increased in cases of increased subcutaneous fluid (increased edema grade), which show high echogenicity as a whole.
Moreover, factors such as the angle of insonation, transducer pressure, and variations in tissue composition can also contribute to inconsistencies in raw brightness levels, further complicating the diagnostic process.
>> (A) Thank you for good advice. A 10 MHz ultrasound linear probe was placed on the skin of the lower leg at a definite point, at 6cm from the upper part of the medial malleolus and 1cm inside the anterior border of the tibia. This probe was suitable to measure the skin. We added the following sentence in Materials and Methods: The same trained evaluator (who had been trained for three months) assessed the measured values under the appropriate conditions. The probe was placed with very, very weak pressure.
As well, numerous environmental and software factors, such as ISO control and ambient lighting, can influence brightness levels. Relying on this data for diagnoses is overly simplistic and could prove dangerous.
>>(A) Yes, we added the following sentence in Materials and Methods: Light output of ultrasound device was fixed from the beginning to the end. It is very important operation.
In addition to the aforementioned concerns, the current methodology's simplicity also limits its applicability and adaptability to different clinical scenarios and patient populations. Leg skin edema can result from various underlying conditions, such as venous insufficiency, lymphedema, or heart failure, which may present with distinct ultrasound image characteristics. A diagnostic method that only considers raw brightness values lacks the capability to differentiate between these conditions or account for patient-specific factors that could influence image interpretation, such as age, body mass index, or comorbidities.
>>(A) Thank you for good opinion. we added the following sentence in Discussion: Our data were based on all selected pregnant women with or without leg edema. Regardless of age, sex, and weight, the same might be true of other leg edemas (only C3 edemas) due to water leakage except for lymphedema, ulcer formation and sclerosis.
Furthermore, the authors' claim that "One of our authors is Dr.Sakamoto, who is a electronic systems engineering. He checked our technique using grayscale of ultrasound photograph and said no problem." does not provide a satisfactory resolution to the concerns I raised. This statement constitutes an unsound appeal to authority instead of a well-reasoned scientific response.
>>(A) We are sorry. We added the following sentence in Discussion: Strictly, water shows very low echogenicity and decreased brightness. These phenomena are seemingly contradictory. We hypothesize that the increased echogenicity of adipose tissue goes beyond low echogenicity of water. Adipose tissue might become firm and its acoustic impedance increases, due to pressure of increased fluid.